# miRNAs as Influencers of Cell–Cell Communication in Tumor Microenvironment

**DOI:** 10.3390/cells9010220

**Published:** 2020-01-15

**Authors:** Ilaria Conti, Gabriele Varano, Carolina Simioni, Ilaria Laface, Daniela Milani, Erika Rimondi, Luca M. Neri

**Affiliations:** 1Department of Morphology, Surgery and Experimental Medicine, University of Ferrara, 44121 Ferrara, Italy; ilaria.conti@unife.it (I.C.); gabriele.varano@unife.it (G.V.); carolina.simioni@unife.it (C.S.); ilaria.laface@unife.it (I.L.); daniela.milani@unife.it (D.M.); erika.rimondi@unife.it (E.R.); 2LTTA—Electron Microscopy Center, University of Ferrara, 44121 Ferrara, Italy

**Keywords:** microRNA, tumor microenvironment, cell–cell communication, extracellular miRNA, cancer therapy

## Abstract

microRNAs (miRNAs) are small noncoding RNAs that regulate gene expression at the posttranscriptional level, inducing the degradation of the target mRNA or translational repression. MiRNAs are involved in the control of a multiplicity of biological processes, and their absence or altered expression has been associated with a variety of human diseases, including cancer. Recently, extracellular miRNAs (ECmiRNAs) have been described as mediators of intercellular communication in multiple contexts, including tumor microenvironment. Cancer cells cooperate with stromal cells and elements of the extracellular matrix (ECM) to establish a comfortable niche to grow, to evade the immune system, and to expand. Within the tumor microenvironment, cells release ECmiRNAs and other factors in order to influence and hijack the physiological processes of surrounding cells, fostering tumor progression. Here, we discuss the role of miRNAs in the pathogenesis of multicomplex diseases, such as Alzheimer’s disease, obesity, and cancer, focusing on the contribution of both intracellular miRNAs, and of released ECmiRNAs in the establishment and development of cancer niche. We also review growing evidence suggesting the use of miRNAs as novel targets or potential tools for therapeutic applications.

## 1. Introduction

Since the discovery of the double-helix structure of DNA in 1953 by Watson and Crick, the study of this genetic material and the processes that it regulates became a central question in the scientific community, resulting in an exponential growth of data and focused on understanding how the genetic code was “interpreted” by the cell and translated into actual biological activity. This led to the definition of what is known as the “central dogma of molecular biology”, describing a linear decodification of the information included in the DNA through RNA transcription first, which will then be translated into proteins [1]. RNA has long been considered only an unstable messenger for short-term storage of genetic information, with no other defined functions. Nevertheless, the coding genome representing only 2% of total DNA and the remaining apparently “useless” DNA was labeled as “junk DNA” without any biological purpose [2].

These theories were totally revolutionized in 1993 by the discovery of a small noncoding RNA, which regulates gene expression at the posttranscriptional level. In the nematode *Caenorhabditis elegans,* the gene *lin-4* was found to codify for a 22-nucleotides long RNA, which had antisense complementarity in the 3′ UTR of lin-14 mRNA. The binding of lin-4 to the complementary mRNA decreased lin-14 protein levels, resulting in a faster progression from the L1 to L2 larval stage [3]. Subsequent studies in *C. Elegans* allowed to identify another 21-nucleotides long RNA (*let-7*) which controls the transition from L4 larval stage to adult, repressing the expression of lin-41 protein [4]. Notably, homologues of *let-7* gene are conserved in many other animal species [5], even though fewer organisms maintained its function as regulator of cell-cycle progression. *Let-7* was required in *Drosophila Melanogaster* for the development of neuromuscular junctions from the larval to the adult stage [6], whereas upregulation of *let-7* promoted differentiation of mouse embryonic stem cells [7]. Due to their abundance, expression patterns and evolutionary conservation, *lin-4* and *let-7* RNAs were included as members of a large regulatory class of noncoding RNAs, 18–24 nucleotides long, called microRNAs (miRNAs) [8]. The microRNAs have been formally recognized as a class of noncoding RNAs in late 2001 [9] and were functionally distinguished from the short interfering RNAs (siRNAs), that directly cleave the target mRNA during the RNA interference process (RNAi) [10].

Since their discovery, multiple studies about miRNAs have been carried out to understand and characterize their biogenesis, mechanisms of action, and regulatory functions in different organisms, from plants to animals [11,12]. Identified miRNA sequences, annotation, and target prediction information are collected in the miRNA database called *miRBase* [13,14,15,16,17]. The miRNA registry was set up in 2002, and the current release of October 2018 (*miRBase 22*) contains 38,589 hairpin precursors and 48,860 mature microRNAs, from 271 organisms [18].

The primary function of miRNAs is to interfere with the translation of target mRNAs or to promote destabilization and degradation of their targets. The activity of miRNAs is context dependent, showing species-specific as well as tissue-specific expression and activities [19]. Left–right asymmetry during neuronal development, cardiogenesis, and fluid balance are some of the biological functions in which they are actively involved [20,21,22]. Alterations in miRNA pathways have been implicated in a variety of human cancers and in infectious and noninfectious diseases, including autoimmune diseases, metabolic disorders, and genetic diseases [20,21,22,23].

Recently, it has been shown that miRNAs can be released in the extracellular circulation and blood (extracellular miRNAs (ECmiRNAs)), either enclosed into membrane vesicles or bound to proteins, preserving them from RNAse-mediated degradation [24,25]. ECmiRNAs can be internalized by target cells and can act as “intercellular” signaling molecules to modulate cellular functions, coordinating tissue development, remodeling, and homeostasis [26,27].

Cancer is a heterogeneous ecosystem where a multiplicity of cell types (including fibroblasts, endothelial cells, and macrophages) and components of the extracellular matrix (ECM), proteases, and cytokines coexist and support the growth and expansion of malignant cells, constituting what is defined as tumor microenvironment [28].

It has been proposed that ECmiRNAs might be involved in the organization of tumor microenvironment, where they can act as mediators of cell–cell communication, promoting constant “interactions” and delivering signals to surrounding cells [29].

In this view, miRNAs can be used by cancer cells to alter and shape tumor niches to satisfy their metabolic and biological needs and miRNAs can be used by cancer cells to alter and shape tumor niches to satisfy their metabolic and biological needs, to efficiently grow and expand, and to eventually migrate and invade distant sites [30].

In this review, we will describe the role of miRNAs in cell–cell communication with a particular focus on the contribution of miRNAs-mediated intercellular signalling to the establishment and remodeling of the tumor microenvironment.

## 2. Biogenesis and Regulatory Functions

MiRNAs are produced from both introns and exons of several genes widely distributed within the genome. About 50% of currently identified miRNAs result from non-protein coding transcripts and are regulated by their own promoters. The remaining miRNAs originate from intragenic regions, mostly introns, and relatively few from noncoding exons; they are transcribed together with their host genes and processed separately [31].

Clusters of miRNA expressed as a single polycistronic transcript have been described as an additional family of miRNAs [32]. Additionally, paralogues of the main cluster are distributed in the genome resulting in the amplification of the same cluster (increasing the number of the miRNAs within each family). The miR-17/92 is one of the most investigated miRNA cluster, which includes six precursors (miR-17, miR-18, miR-19, miR-19b, miR-20a, and miR-92a) within the locus of the noncoding gene MIR17HG, on chromosome 13 [33]. Two paralogues of miR-17/92 cluster, namely miR-106b/25 and miR-106a/363, are located respectively on chromosomes 7 and X, adding 9 miRNAs to the family, for a total of 15 miRNAs [34].

MiRNA biogenesis begins with the transcription of a primary miRNA (pri-miRNA) by RNA polymerase II in the nucleus. The pri-miRNAs are generally several hundred nucleotides long with a 5′ guanosine-cap and a 3′ polyadenylated tail. The long sequence is then processed into a 70–120 nucleotide long precursor RNA (pre-miRNA) by the Microprocessor complex [35]. This multi-protein complex consists of two molecules of DGCR8 (Di George Syndrome Critical Region 8), a double-stranded RNA (dsRNA)-binding protein also known as Pasha, and one molecule of the RNAse III enzyme called Drosha. The Microprocessor complex is able to recognize and cleave the stem-loop portion of the pri-miRNA to generate a 2 nucleotides overhang at the 3′ end of the pre-miRNA [36]. The newly miRNA precursor is exported out of the nucleus into the cytoplasm by the nuclear transport receptor protein Exportin 5 (XPO-5)/Ran GTP complex [37]. In the cytoplasm, the RNAse III endonuclease Dicer, together with the Protein Kinase RNA-activated (PKR) and the two dsRNA-binding proteins TRBP and PACT (respectively, HIV Trans-activation Response RNA Binding Protein and Protein Activator of PKR) process the pre-miRNA, removing the terminal loop to obtain a double-stranded 18-23 nucleotide long mature miRNA [38,39]. MiRNA biogenesis depends on the expression of both Drosha and Dicer during pre-miRNA processing, while XPO-5 is dispensable for the generation of mature miRNAs. Coherently, knockout of Drosha, Dicer, and XPO5 in human HTC116 cell lines decreased cellular proliferation, presumably due to the lack of miRNAs biogenesis. Western-blot and sequencing analyses showed miRNA reductions of 96.5% and 96% compared to the control in the absence of Drosha and Dicer, respectively. On the other hand, XPO-5 deletion resulted in a 29% reduction in the expression of total miRNAs [40].

Both miRNA strands theoretically can be loaded in the RNA-induced silencing complex (RISC) and bind the target mRNA [41]. Thermodynamic asymmetry and stability of base pairing at 5′ ends of the duplex allowed to separate the two strands and to determine their role as guide (-5p, from the 5′ arm) or passenger (-3p, from the 3′ arm) strands. Generally, the guide strand has lower 5′ stability or an uracil in position 1 [42]. The passenger or -3p miRNA strand is usually degraded, while the guide strand is loaded into one of the catalytic Argonaute family proteins (AGO1-4 in humans), in an ATP-dependent manner [41]. Furthermore, many -3p miRNAs have important roles in physiology and pathophysiology, such as miR-455-3p which targets the 3′ UTR of PAK2 (P21-Activated Kinase 2), inhibiting the cartilage degeneration pathway in human osteoarthritis models both in in vitro and ex vivo experiments [43]. The mature miRNA-AGO complex associates with other proteins (including GW182, PABPC (Poly(A)-Binding Protein C), and two poly(A)deadenylase complexes PAN2/PAN3 and CCR4-NOT), constituting the RISC complex, to eventually induce the repression of the target mRNAs [44,45].

Noncanonical miRNA biogenesis pathways have been also described, accounting for the residual expression of miRNAs (e.g., mirtrons, m7G-capped pre-miRNAs, and shRNAs) upon Drosha, Dicer, or XPO-5 knockout. Through these pathways, mirtrons (miRNAs originated from intron splicing) and 7-methylguanosine(m7G)-capped pre-miRNAs are produced independently of the Drosha/DGCR8 cleavage [46,47,48] while short hairpin RNAs (shRNAs) are generated in absence of Dicer activity [49].

Lack of the proteins involved in miRNAs production has been recently correlated to several biological processes, such as organ development, RNA processing, and regulation of transcription or genome integrity [50]. Loss of stem cell properties and premature differentiation in murine forebrain neural progenitors was induced by the stabilization of the neurogenic transcription factor Neurogenin 2 (Ngn2) mRNA in the absence of Drosha [51]. In another study, in a mouse model of medulloblastoma, Dicer depletion resulted in the accumulation of DNA damage and increased cell death [52].

The degree of complementarity between the miRNA and target mRNA determines whether translational repression or mRNA degradation will occur. MiRNAs recognize and anneal to specific sequences located at the 5′ or 3′ UTR of target mRNAs, called MiRNA Response Elements (MREs). The larger the miRNA:MRE complementarity, the higher the probability that mRNA cleavage will occur; a complete miRNA:MRE interaction causes degradation of the target mRNA mediated by the activity of AGO2 endonuclease [53]. Within the RISC complex, the poly(A)deadenylases PAN2/PAN3 (poly(A)-nuclease 2/poly(A)-nuclease 3) and CCR4-NOT (Carbon Catabolite Repressor 4-Negative Regulator of Transcription) are responsible of deadenylation of the target mRNA in the presence of Poly(A)-Binding Protein C (PABPC) [54]. Then, the Decapping Protein 2 (DCP2) removes the m7G-cap, exposing the target strand for Exoribonuclease 1 (XRN1)-dependent 5′-3′ degradation [55].

On the other hand, a partial miRNA:MRE annealing due to the presence of mismatches within the guide miRNA prevents AGO2 endonuclease activity. The miRNA:AGO2 complex acts as a repressor of translation inhibiting the binding and activity of ribosome on target mRNA, but the mechanism is not clearly defined yet [56]. In an in vitro RNA interference (RNAi) assay, a 4-nucleotide mismatch in the complementary region between lin-41 mRNA and let-7 siRNA, was sufficient to prevent degradation of the target mRNA in the presence of a *Drosophila* embryo lysate [57].

Moreover, the decision between degradation and repression of translation of the target mRNAs not only depends on the strength of the interaction between the target mRNA and the miRNAs but also is influenced by the composition of the RISC complex [58]. Depletion of CCR4-NOT or of the decapping complexes inhibited mRNA degradation but not translational repression in *Drosophila* [59]. Finally, expression of the target mRNAs, as well as the abundance of miRNAs, are influenced by the identity of the cell and its metabolic and activation status, both important determinants of miRNA-mediated gene regulation [60,61].

Although miRNAs are identified as negative regulators of gene expression, some studies have shown their role as translational activators [62]. They have been recently involved in a process called RNA activation (RNAa), where RNA polymerase II, together with other transcription factors, such as RNA Helicase (RHA) and heterogeneous nuclear Ribonucleoproteins (hnRNps), are recruited on the gene promoter through the interaction with miRNAs, boosting the transcription of the gene [63]. However, it has been proposed that this dual function of miRNAs is influenced by their localization within cellular compartments, as the AGO2-miRNA complex can translocate to the nucleus through the interaction with Importin 8 [64]. MiR-24-1 was detected both in the nucleus and in the cytoplasm of miR-24-1-transfected HEK293T cells by Northern Blotting analysis, where it showed different activities: it acted as a canonical repressor of gene expression in the cytoplasm, whereas nuclear miR-24-1 altered histone modification and increased the recruitment of transcription factors (such as p300 and RNA polymerase II) on enhancers and promoters of its target genes (e.g., FBP1 (Fructose-1,6-Biphosphatase 1), LSMD1 (LSM Domain-containing protein 1), CYB5D1 (Cytochrome B5 Domain containing 1), and KDM6B Lysine demethylase 6B), activating their expression [65]. Although further investigation is required to disclose its mechanisms of action, RNAa represents an emerging field, which could have important implications during tumor development and progression.

Upon their tissue-specificity and, more recently, on their dual effect on gene expression, miRNAs have been identified as important regulators of many biological processes. Stem cell proliferation and differentiation, organ development, cancer, and infectious diseases have been shown to be influenced and regulated by various miRNAs [66]. *Let-7* miRNA and the miR-290-295 cluster have opposite effects on stem cell biology in mice, inhibiting or promoting stem cell renewal, respectively: *let-7* was detected to favor differentiation of stem cells and to antagonize the action of the miR-290-295 cluster, whereas the latter facilitates G1-S transition to maintain embryonic stem cells (ESCs) in a pluripotent state [67]. Indeed, transcription factors critical for the ESC pluripotent state, such as Oct3/4 (Octamer Binding Transcription factor 3/4), Nanog, and Sox2 (Sex determining region Y-box 2), were shown to increase the expression of the miR-290-295 cluster by binding its promoter [68]. Given its role in regulating the ESC cell cycle, miR-290-295 cluster has been called “ESCC family of miRNAs” [69].

As mentioned above, miRNAs are responsible for the development of different tissues and organs, including the nervous system, where the temporal and spatial control of miRNAs expressions is critical, considering that a single cellular progenitor potentially gives rise to a variety of cell identities [70]. Of the 351 miRNAs that were identified in primary cultures of neurons, astrocytes, oligodendrocytes, and microglia, 116 miRNAs were differently expressed among those different populations [71]. In *C. elegans*, the asymmetric expression of *lys-6* miRNA in two chemosensory neurons (ASEs) was required for the left/right production of guanylyl cyclase genes and the functional lateralization of ASE neurons [72]. *Lys-6* miRNA was identified as responsible for the left chemosensory neuron called ASE Left (ASEL), while its absence in the opposite symmetry allowed the development of the ASE Right (ASER) chemosensory neuron. In the right symmetry of the nematode, miR-273 downregulated the level of *lys-6* by binding the 3′ UTR of its transcription factor Die-1 (Dorsal intercalation and elongation defective-1) [73].

Some miRNAs show a temporal restricted expression. MiR-9 and miR-124 are specifically expressed during early neurogenesis, where they promote the differentiation of neural stem cells and the development of dendritic branching inhibiting the expression of the Ras-related protein Rap2 [74,75]. In some cases, they have been involved in the regulation of cell fate. The miR-9–miR-124 combination can convert human fibroblast into functional neurons as a result of epigenetic alterations silencing the expression of BAF53a [76]. In other cases, miRNAs expression was restricted to highly specialized compartments. For example, miR-134 is involved in the regulation of the size of the dendritic spine size by locally inhibiting LIMK1 (LIM domain Kinase 1), a kinase responsible of actin polymerization [77].

Finally, miRNAs can be expressed according to specific cell states. P53 is a transcription factor that acts as safe-keeper of cellular integrity, inducing cell cycle arrest or apoptosis, in response to cellular stress, such as DNA damage, oncogene activation, and hypoxia [78]. Some miRNAs have been recently identified to activate p53 through the downregulation of its negative regulators. Specifically, miR-192/194/251, miR-143/145, miR-29b, miR-605, miR-25, miR-32, miR-18b, and miR-339-5p repress MDM2 (Mouse Double Minute homolog 2), a key regulator of p53, inducing p53 activation, growth arrest, and apoptosis [79]. Also, interference with MDM4 (Mouse Double Minute homolog 4) expression, another p53 negative regulator, by miR-10a, miR-191-5p, miR-887, miR-661, miR-34a, miR-199a-3p, and *let-7* results in p53 activation and cell death [80]. On the other end, p53 itself can regulate the expression of specific miRNAs through the activation of their transcription in the presence of DNA damage, inducing cell cycle arrest, apoptosis, and senescence [81]. MiR-192, miR-194, miR-215, miR-145, miR-107, and the family miR-34 are transcriptional targets of p53 that have reduced expression levels in cancers [78]. The members of miR-34 family (miR-34a and miR-34b/c) interfered with the expression of Cyclin E2, Cyclin-Dependent Kinase 4 and 6 (CDK4 and CDK6), and Bcl2 (B-cell lymphoma 2), inducing cell-cycle arrest in both cancer-derived primary cultures and cell lines [82,83]. MiR-145 targets the proto-oncogene c-Myc mRNA, inhibiting the tumor growth both in vitro and in vivo [84]. MiR-192 and miR-215 are involved in cell-cycle regulation, downregulating genes responsible for DNA synthesis (e.g., ERCC3 (Excision Repair 3) and LMNB2 (Lamin B2)) and cell-cycle checkpoints (e.g., CDC7 (Cell Division Cycle 7) and MAD2L1 (Mitotic Arrest Deficient 2 Like 1)), as determined by gene expression profiling of miR-192-215 transfected HCT116 DICERex5 cells (Human Colorectal Cell Line with reduced DICER function) [80]. Less frequently, p53 was also reported as a repressor of certain miRNAs, such as miR-224, miR-502, and the miR-17/92 cluster, resulting in lower cell proliferation (miR-224) [85] and increased apoptosis (miR-17/92 cluster) [86].

Recently, miRNAs have been identified within mitochondria and named “mitomiRs” due to their cellular localization [87]. They participate in the regulation of mitochondrial functions, acting both on mitochondrial mRNAs and on mitochondrial DNA (mtDNA) transcription (e.g., fission/fusion events; mitophagy, i.e., removal of damaged mitochondria; and mitochondrial oxidative phosphorylation OXPHOS) and their alterations contribute to cellular energetic instability, leading to the development of diseases such as metabolic syndrome [88]. MitomiRs at both the mature and precursor states have been evaluated within mitochondria, raising different hypotheses on their origin: miRNAs/pre-miRNAs transcribed from the nuclear DNA are translocated within the mitochondria where the miRNA biogenesis is completed, or they could be directly transcribed from the mtDNA, but the last one still remains a hypothesis [87]. Concerning miRNAs translocation to mytochondria, the cooperation between the cytosolic AGO2 and the Polynucleotide Phosphorylase (PNPase), a protein of the inner mitochondrial membrane, allowed the import of miR-378 within these cellular organelles in HL-1 cells [89]. Inside the mitochondrion, miR-378 bound ATP synthase 6 (ATP6) mRNA, decreasing its synthase activity both in vitro and in vivo [89] and contributing to the development of type 2 diabetes [90]. On the other hand, the complex AGO2-miR-2392 silenced the transcription of mitochondrial genome by binding mtDNA instead of the promoter contributing to tumor metabolism [91].

Eukaryotic miRNAs (host miRNAs) are involved and influence virus–host interaction, promoting replication and propagation of viruses. For example, liver-specific miR-122 was shown to stimulate protein translation [92], genome replication, and packaging [93] in addition to a genomic protective activity upon recognition and binding to 5′ UTR of the Hepatitis C Virus (HCV) RNA [94]. However, some viruses (such as Epstein Barr Virus, Simian Virus 40, and Polyomaviruses) generate their own miRNAs to promote specific biological processes in the host cell, such as apoptosis and/or evasion from the immune response, to successfully complete their cycle [95,96]. From an evolutionary perspective, the generation of miRNAs is more efficient and rapid than the production of new proteins with the same functions. Indeed, Human Cytomegalovirus (HCMV) is able to express miR-UL112-1, miR-US5-1, and miR-US5-2 viral-miRNAs (v-miRNAs) upon transduction of Normal Human Dermal Fibroblast (NHDF) and HeLa cell lines. These v-miRNAs caused the reorganization of the cellular secretory pathways (e.g., the loss of Golgi) and repressed the mRNAs of IL-6 (Interleukin 6) and TNFα (Tumor Necrosis Factor α) cytokines blocking host-mediated immune responses [97].

The discovery of miRNAs allowed improving our knowledge on a multiplicity of biological processes, increasing the complexity of the mechanisms driving cellular biology. Given their involvement in the regulation of biological processes both in physiologic and pathologic conditions, they constitute possible targets for the development of specific therapeutic strategies. Among them, interference with v-miRNAs may represent a promising therapeutic approach as an innovative antiviral therapy. In this view, v-miRNAs are temporarily expressed, and their manipulation may not cause cellular abnormalities that a host miRNA’s manipulation may cause, thus reducing cytotoxicity and off-target effects in noninfected cells [96,98].

## 3. miRNA: A Factor in Multifactorial Diseases

MiRNAs are involved in the regulation of a multiplicity of biological processes, and conditions promoting their aberrant expression—i.e., amplification or deletion of miRNAs genes, altered transcriptional controls, defects in the miRNAs’ biogenesis, and epigenetic changes—have been associated with the development of various disorders and pathogenic conditions, from metabolic diseases to cancers. Over 2000 human mature miRNAs described in the last version of *miRBase* have been identified, through in silico high-throughput analysis of the total *miRBase* entries followed by Microarray and Northern-Blotting analyses of selected putative miRNAs transfected cells (e.g., HeLa and HEK293T cell lines) [99]. MiRNAs are both pleiotropic and redundant, and it has been proposed that at least 30% of human genes are regulated by the cooperation among miRNAs: one mRNA can be recognize by different miRNAs and one miRNA can recognize different mRNAs [100]. This interplay complicates the association of a single miRNA to a specific pathologic effect, as multiple miRNAs redundantly reinforce the same cellular fate [101]. Instead, the identification of disease-specific groups of miRNAs, as well as their pattern of expression, may represent a promising strategy with strong diagnostic and prognostic potential [100].

Here, we will discuss the miRNA involvement within some complex diseases, including Alzheimer’s disease, obesity, and cancer.

**Alzheimer’s Disease (AD)** is a neurodegenerative disorder characterized by the presence of two neuropathological features, namely the senile plaques, consisting of β-Amyloid peptides (Aβ), and the Neurofibrillary Tangles (NFTs), that contain highly phosphorylated microtubule-associated protein Tau [102]. The etiology and the molecular mechanisms driving AD pathogenesis are still poorly known (excess Aβ production or inefficient Aβ clearance and formation of NFT) [103]. Several studies support the hypothesis where the accumulation of Aβ leads to the generation of toxic aggregates, which results in neuronal dysfunction and, ultimately, cell death [102].

Several miRNAs have been described to be involved at different stages of AD development, from the regulation of the Aβ or Tau protein production to their clearance. MiR-106a/520c, miR-20, miR-101, miR-16, and miR-153 can bind the 3′ UTR of Amyloid Precursor Proteins (APPs), resulting in their lower expression, both in vitro and in vivo [103]. Alternatively, miR-29-a/b-1 and miR-29c were found to regulate Beta-Secretase Enzyme 1 (BACE1), the protease responsible for APP digestion [104]. Decreased APP and BACE1 expression reduce Aβ production, decreasing the probability of plaque deposition. On the other hand, Aβ deposition can be reduced by mechanisms of clearance of Aβ plaque (e.g., Triggering Receptor Expressed on Myeloid cells (TREM) and endosomal-lysosomal enzymes). The myeloid/microglial cell surface amyloid sensor-receptor (TREM2) and the endosomal-lysosomal system enzymes are implicated in the recognition and degradation of amyloid proteins and debris, preventing the deposition of Aβ plaque; in AD patients, they are negatively regulated by miR-34a and miR-128, respectively [105,106]. However, other diseases could also be derived by altered Aβ clearance mechanisms. A correlation between AD and atherosclerosis has been established upon induction of brain vessel damage as a consequence of Aβ accumulation [107].

The imbalance between Tau phosphorylation and dephosphorylation has been proposed as an alternative mechanism underlying the generation of senile plaques and the development of AD [102]. MiR-125b, miR-132, miR-26, and miR-146a indirectly trigger the hyper-phosphorylation of Tau protein acting on different cellular factors. For example, increased expression of miR-125b activated Cyclin-Dependent Kinase 5 (CDK5) and p35/25, inducing apoptosis and Tau hyper-phosphorylation in neuron cells [108]. Alternatively, miR-146a inhibited the expression of Rho-associated coiled-coil kinase 1 (ROCK1), promoting Tau phosphorylation both in in vitro and in vivo experiments: functional ROCK1 activates PTEN (Phosphatase and Tensin homolog), which in turn dephosphorylates Tau [109].

Mitochondrial activities and, in particular, their alteration have been suggested to promote the development of AD considering the mitochondrial metabolism and cellular respiration: an AD brain changes its metabolism, preferring the use of lipids and amino acids as energetic sources instead of glucose due to its low level, and mitochondria are the major producer of ROS (Reactive Oxygen Species) that can contribute to brain damage [88,110]. MiR-330 overexpression in an AD mouse model inhibited the activation of MAPK pathway (Mitogen-Activated Protein Kinase) by binding the 3′ UTR (untranslated region) of VAV1 (Vav guanine nucleotide exchange factor 1), reducing mitochondrial dysfunction, oxidative stress, and Aβ plaque deposition [111].

Finally, expression of several circulating miRNAs was altered in blood from AD patients. Specifically, circulating miR-26b, miR-30e, miR-34a, miR-34c, miR-485, miR-200c, miR-210, miR-146a, and miR-125b were downregulated both in the brain and blood of AD patients compared to controls, suggesting a role for these miRNAs as potential markers for Alzheimer’s disease [112]. Accordingly, expression of miR-125b is associated with the Mini Mental State Examination (MMSE), which is the most common tool for the evaluation of AD symptoms [113].

**Obesity** is a multifactorial disease, and over 30% of the population worldwide is nowadays considered overweight or obese [114]. Calorie intake, lifestyle, and genetic predisposition are all factors involved in the development of obesity, and the altered equilibrium between energy intake and consumption may lead to abnormal white fat accumulation [115]. Low-Density Lipoproteins (LDL) and High-Density Lipoproteins (HDL) contribute to lipid and cholesterol metabolism, and their production can be regulated by miRNAs. MiR-122, miR-148a, miR-128-1, and miR-30c have been identified as important regulators of plasma levels of LDL, cholesterol biosynthesis, and expression of hepatic LDL Receptor (LDLR). Specifically, miR-148a or miR-128-1 inhibition in the HepG2 liver cancer cell line increased the expression of ATP-Binding Cassette A1 (ABCA1), LDLR, and plasma HDL, while they reduced plasma LDL levels [116]. Several other miRNAs are involved in HDL metabolism acting on ABCA1 (miR-33, miR-758, miR-144, miR-26, miR-27a/b, miR-302a, miR-148a, miR-128-1, and miR-19b), HDL Scavenger Receptor class B type 1 (SR-B1; miR-455, miR-125a, miR-185, miR-96, and miR-233), Angiopoietin Like 3 (ANGPTL3), and Glycerol-3-Phosphate Acyltransferase (GPAM; miR-27) [117].

Obesity is generally characterized by the accumulation of white fat at the expenses of brown fat cells. More recently, the beige or *brite* fat cells have been identified as a new type of adipocyte [118]. They can be distinguished from the other two fat cells (white and brown adipocytes) based on their transcriptional program and on the expression of specific markers, such as T-box1, tumor necrosis factor receptor member 9 (TNFRSF9 also named CD137), and Transmembrane protein 26 (Tmem26). Also, they express brown genes involved in thermogenesis, such as the Uncoupling Protein 1 (UCP1) [119]. The origin of *brite* is still unclear, and two different hypothesis have been proposed, suggesting that they are generated in response to diverse environmental or cellular cues (low temperatures; physical activity; thyroid hormones; nutritional or pharmaceutical, e.g., β3-adrenergic, compounds) upon i) de novo differentiation from progenitors cells or ii) transdifferentiation from white mature adipocytes [120,121]. MiR-196a, miR-26, and miR-30 promote the expression of beige-specific markers, whereas miR-155, miR-133, miR-27b, and miR-34 antagonize the transition towards *brite* cells (“browning process”), indirectly inhibiting UCP1 expression (the transcription factor PR-Domain containing 16, PRDM16, which in turn induces UCP1 expression, was repressed in muscle cells overexpressing miR-133), whereas its expression was induced in miR-27b knockout mice [122,123]. Studies have recently shown the involvement of mitochondria in adipocyte metabolism, especially in the browning process [124]. MiR-26 family is identified as a pivotal factor to induce energy dissipation pathways. In transfected hMADS cells (human Multipotent-Adipose Derived Stem cells) miR-26a increased the expression of UCP1 and induced changes toward the mitochondrial brown adipocytes morphology by targeting the 3′ UTR of ADAM17 (ADAM Metallopeptidase Domain 17) [125].

Pharmacological treatment and dietary strategies aimed at promoting the transition from white to beige fat cells have been considered to fight obesity. β3-adrenergic agonists and dietary-derived molecules such as resveratrol (red wine) and capsaicin (spicy pepper) are able to activate the expression of thermogenic genes in white adipocytes, resulting in the thermal conversion of lipids and their conversion into brown/beige cells [126].

**Cancer** is a complex disease derived from the cooperation of several factors, including genetic and environmental components. The dysregulation of different cellular pathways are involved in the development of this disease, such as the PI3K/Akt/mTOR pathway (Phospoinositide 3-Kinase/Serine-Threonine protein Kinase/mammalian Target Of Rapamycin), promoting uncontrolled cellular proliferation and tumor growth and leading to physical and metabolic exhaustion of the host until its death [127]. The PI3K/Akt/mTOR signalling pathway regulates several cellular processes, including cell proliferation and cell-cycle progression, survival, cell metabolism, and transcription [101]. PTEN is one of the main negative regulators of this pathway, mediating the dephosphorylation of Phosphatidylinositol (3,4,5)-trisphosphate (PtdIns (3,4,5)P3 or PIP3) to PIP2 and preventing the activation of the pathway [128]. In this view, PTEN acts as a tumor suppressor, preventing PI3K/Akt/mTOR activation, and is highly mutated in a variety of human cancers, including breast cancer, melanoma, and lung and prostate carcinoma [127,129]. Cancer-associated PTEN mutations are loss of function mutations, leading to the upregulation of the PI3K/Akt/mTOR pathway, abnormal cell proliferation, and metabolic changes [130]. The miR-181 family is a cluster of 6 members (miR-181a1, miR-181a2, miR-181b1, miR-181b2, miR-181c, and miR-181d) that regulates PTEN in immune cells, reprogramming their metabolism and interfering with their differentiation. Specifically, thymocytes from miR-181a1/b1-deficient mice showed increased PTEN levels and a reduction in PI3K signalling results in a block of NKT-cells production [131]. MiR-181a1/b1-miR-181a2/b2 double knockout mice showed reduced overall survival, while the miR-181a1/b1-miR-181a2/b2-miR-181c/d triple knockout mouse was embryonically lethal, suggesting a certain degree of redundancy among the different members of this family, where each member can compensate for the absence of the others, and pointing to an essential role of the miR-181 family during differentiation and embryonic development [132].

Several miRNAs, in addition to the miR-181 family, are involved both in the development and in the differentiation process of Acute Lymphoblastic Leukemia (ALL), a pediatric malignant disorder. The miR-150, miR-155, and miR-17/92 cluster have been identified as essential regulators of hematopoiesis, influencing maturation of lymphoid precursors. Their aberrant expression alter lymphopoiesis and participate to the development of ALL [133]. Indeed, a characteristic pattern of under- or overexpression of miR-126, miR-128, miR-146b, miR-148, miR-151, miR-191, miR-424, miR-425-5p, and miR-629 was associated with the commitment of the ALL towards either B or T cell lineages [134]. Moreover, in cases of ALL, the expression pattern of specific miRNAs has been associated not only to the diagnosis but also to the prognosis and response to therapy. For example, miR-92a overexpression correlated with worse outcomes [135] while high miR-128b was associated with a better prognosis and response to corticosteroid treatments (Prednisolone) [136].

The involvement of miRNAs in human cancer was firstly showed in B-cell chronic lymphocytic leukemia (B-CLL). The 13q14 deletion, which is the most common genetic alteration in B-CLL (detected in over 50% of cases) impasses a gene encoding for the miR-15a/miR-16-1 cluster of miRNA [137]. These miRNAs have been described to have a tumor-suppressor function, being able to repress the expression and activity of several oncogenes, including Bcl2 and Cyclin D1 [138]. In addition to B-CLL, miR-15a/miR-16-1 are also deleted or mutated (loss of function mutations) in a variety of cancers (e.g., melanoma, colorectal cancer, and bladder cancer) [138,139].

As previously mentioned, miRNAs are involved in the regulation and the control of several biological processes, including cellular metabolism and energy biosynthesis, which are particularly relevant during malignant transformation and tumor progression. Indeed, cancer cells shape the activity of specific miRNAs to support their metabolic needs in order to proliferate and metastasize. Specifically, miR-200 family, miR-122, miR-17/92 cluster, miR-15a/16-1, miR-29, miR-326, and miR-133 target different glycolytic enzymes and are used by cancer cells to rewire their metabolic status, switching from aerobic respiration towards anaerobic glycolysis and allowing a massive production of energy and of the required intermediates and metabolites (e.g., amino acids, nucleotides, and lipids). This phenomenon is known as “Warburg effect” [140]. For example, miR-200 family (miR-200a, miR-200b, and miR-200c) regulates the Phosphoglucose Isomerase/Autocrine Motility Factor (PGI/AMF), which is one of the first enzymes of the glycolytic pathway. The repression of these miRNAs results in the overexpression of the glycolytic enzymes and increased metastasis in MDA-MB-231 breast cancer cells. PGI/AMF induced Epithelial-Mesenchymal Transition (EMT), increasing the expression of ZEB1/ZEB2 (mesenchymal markers; Zinc finger E-Box-binding homeobox 1/Zinc finger E-Box-binding homeobox 2) through the Nuclear Factor-kB (NF-kB) in breast cancer cells [141]. Similarly, downregulation of miR-200 family (miR-200b and miR-200c) and miR-205 was responsible of the EMT in aggressive pancreatic cancer cells by increasing the expression of ZEB1/ZEB2 [142].

Since cancer cells prefer glycolysis for a major energy and metabolites production, the physiological mitochondrion functions (i.e., tricarboxylic acid (TCA) Krebs Cycle, ETC Electron Transport Chain and OXPHOS) are suppressed [124]. 

miR-126 acts as a tumor suppressor downregulating IRS1 (Insulin Receptor Substrate 1) and restoring the mitochondrial TCA metabolism in malignant mesothelioma H28 and Met5A cell lines. As a consequence of targeting the 3′ UTR of IRS1, miR-126 inhibited the activation of Akt, promoting the gluconeogenesis and the glycolysis and reducing the malignant phenotype [143]. 

Mitochondria are also involved in breast and melanoma cancer stem cells. It has very recently been reported that miR-1 targeted the LRPPRC (Leucine-Rich Pentatricopeptide-Repeat containing) protein and MINOS1 (Mitochondrial Inner Membrane Organizing System 1) and GDP2 (Glycerol-3-Phosphate Dehydrogenase 2) genes, destroying mitochondria, which in turn caused block of cell cycle at the G0/G1 phase and the suppression of the tumorigenicity both in in vitro and in vivo experiments [144].

## 4. miRNAs and Their Role in the Tumor Microenvironment

Cancer development is the results of a multistep process where normal cells acquire a combination of genetic events allowing them to survive and expand in an initially nonpermissive environment, where nutrient availability might be insufficient to fulfil the metabolic requirement of the transformed cells and where cancer cells need to evade from immune recognition and clearance. This initially unfavorable environment is progressively shaped and reorganized to achieve a cellular and extracellular composition that allows tumor cells to engraft and rapidly expand, defined as tumor microenvironment [145,146]. This process requires constant interaction with the neighboring cells through physical contact and releasing factor with paracrine activity (cytokines, lipids, and neurotransmitters). T cells, dendritic cells, macrophages, fibroblasts, endothelial cells, adipocytes, and components of the extracellular matrix (ECM) such as proteases and cytokines have been identified as part of the tumor microenvironment. Interestingly, the components of this niche do not have a malignant origin but they are induced by tumor cells and coevolve with them to promote tumorigenesis [29]. The organization of a tumor microenvironment occurs within the primary tumor as well as at distal sites, where metastatic tumor cells need to generate a permissive environment that facilitate their growth [146].

Recent studies have shown the involvement of miRNAs in cell–cell communication. Expression of miRNAs can be induced within target cells by different cellular factors or microenvironmental signals, for example, glucose levels in the ECM prompt glioblastoma cells to either proliferate or migrate away from a nonpermissive environment. Both mathematical models and in vitro experiments showed an increase of miR-451 expression in glioblastoma cells in the presence of glucose. Mechanistically, miR-451 downregulates the AMPK (AMP-activate Protein Kinase) complex, promoting cellular proliferation and attenuating cellular migration [147,148]. On the contrary, the presence of lysophosphatidic acid (LPA) in the ECM promotes metastasis both in vitro and in vivo in models of human breast cancer. LPA activated the G-protein coupled receptor LPAR1, which promotes the upregulation of miR-21 in breast cancer cells through a PI3K/ZEB1-dependent pathway. MiR-21 was responsible for the inhibition of anti-metastatic genes, such as PTEN, PDCD4 (Programmed Cell Death protein 4), and SPRY2 (Sprouty Homolog 2), promoting tumor migration [149].

Numerous cytokines can be detected at various concentration within the tumor microenvironment and can activate signalling pathways resulting in the expression of miRNAs to escape immune surveillance and to promote proliferation, metastasis, and tumor invasion [28]. Tumor-derived IL-6 downregulated the expression of miR-19a-3p in Tumor-Associated Macrophages (TAMs), polarizing them towards an immune-suppressive phenotype (M2-polarization) by increasing the expression of FOS-related antigen 1 (Fra-1) and finally promoting migration and invasion of 4T1 breast cancer cells [150].

Furthermore, same cytokines can act on different components of the same cancer niche, such as Transforming Growth Factor β (TGF-β). Release of TGF-β by colorectal cancer cells led to the expression of miR-491 in CD8+ T cells within the tumor microenvironment, which was able to suppress T lymphocytes activation and proliferation, favoring immune evasion and cancer progression [151]. Myeloid-Derived Suppressor Cells (MDSCs) are cellular components of the tumor microenvironment characterized by immunosuppressive properties. Tumor-derived TGF-β1 induced miR-494 expression in MDSCs in vivo, interfering with PTEN expression and miR-494 and leading to increased proliferation and accumulation of MDSCs within the tumor microenvironment, sustaining tumor invasion of 4T1 murine breast cancer cells [152]. 

During the establishment of the tumor microenvironment, cancer cells can also reprogram fibroblasts towards a “transformed phenotype” to facilitate angiogenesis and metastasis. Within the colorectal cancer niche, the expression of miR-21 in Cancer-Associated Fibroblasts (CAFs) was induced by tumor-derived TGF-β. MiR-21 upregulated α-Smooth Muscle Actin (α-SMAD) expression, leading to transdifferentiation of fibroblasts to myofibroblasts and ultimately favoring cancer metastasis [153]. High migratory and invasive indexes were identified in Scirrhous type gastric cancer following the release of TGF-β. This cytokine induced the expression of miR-143 in CAFs, enhancing collagen type III synthesis and promoting cancer invasion [154].

Finally, several tumors release Granulocyte Macrophage-Colony Stimulating Factor (GM-CSF), which induces the expression of miR-200c in MDSCs. MiR-200c downregulates PTEN and Friend Of Gata 2 (FOG2), promoting PI3K/Akt activation and resulting in accumulation of MDSCs and suppression of immune responses within the tumor microenvironment [155].

Cellular factors that activate miRNAs pathways and the related effects in recipient cells are summarized in Table 1 and Figure 1.

Cellular and extracellular ligands are able to regulate the expression of specific miRNAs in order to boost or decrease the tumorigenesis process. Therefore, altered miRNAs patterns could be used as biomarkers for early detection during cancer development. Moreover, targeting components of the tumor microenvironment required for efficient cell–cell communication might provide the basis for the development of novel therapeutic approaches to be combined to standard therapies.

## 5. Extracellular miRNAs

Although the majority of miRNAs are produced and exert their activities as intracellular molecules within the same cell of origin, a significant number of miRNAs have been recently detected also at the extracellular level. These circulating miRNAs or extracellular miRNAs (ECmiRNAs) are remarkably stable in the extracellular environment despite the presence of high RNAse activity. ECmiRNAs have been evaluated in a variety of biological fluids, such as serum, plasma, tears, saliva, urine, breast milk, peritoneal fluid, bronchial lavage, seminal fluid, follicular fluid, and cerebrospinal fluid [156,157]. Given their easy detection in biological fluids, it has been proposed to use ECmiRNAs as biomarkers with diagnostic and prognostic value [25]. In this view, the detection of ECmiRNAs in liquid biopsies provides a noninvasive tool that allows to monitor their levels over time, as a predictive marker for response to therapy and as a measurement of residual disease [158].

Several studies proposed that the release of ECmiRNAs is the result of a passive cellular mechanism to maintain a stable miRNA:mRNA ratio within the cells. Following this hypothesis, a miRNA concentration higher than its targeted mRNA results in the secretion of vesicles containing miRNAs [159]. Alternatively, recent studies demonstrated a directional delivery of ECmiRNAs towards target cells where, once internalized, they actively participate in the regulation (activation or repression) of a variety of cellular pathways. Vesicles containing miR-143 and miR-145 can be released by endothelial cells to control the differentiation of smooth muscle cells (SMCs) and the promotion of an atheroprotective phenotype of SMCs in mice [160].

Hence, the physiological role of ECmiRNAs as endocrine and paracrine molecules and their involvement in the promotion of different pathologies suggest they are deliberately released by cells.

ECmiRNAs show higher stability compared to synthetic exogenous miRNAs spike-ins, suggesting the existence of a safe-guard system preserving them from the extracellular environment. An increasing number of studies identified different kind of ECmiRNAs: they can be i) enclosed within Extracellular Vesicles (EVs) or ii) bound to proteins. Among the EVs, exosomes, microvesicles, and apoptotic bodies represent the main structures embedding ECmiRNAs. They differ from each other in dimension, membrane composition, and biosynthesis [161,162]. In particular, exosomes are small vesicles of 30–100 nm in diameter that are originated by internalization of late endosomes or multivesicular bodies [163]. Microvesicles are heterogeneous membrane structures of 100–1000 nm in diameter. They are originated by budding and fission of plasma membrane; therefore, their lipid and protein contents depend on the composition of the membrane of origin [164]. Apoptotic bodies are larger vesicles of 1–5 µm in diameter in which budding generally occurs during programmed cell death [165]. Since the membrane blebbing for the generation of apoptotic bodies is a late step of cell death, the loading of miRNAs into these vesicles might occur as a random process. However, more studies are necessary to understand the mechanisms driving the specific and selective release of apoptotic bodies containing miRNAs [166].

Recently, two conserved motifs (GGAG and GGUC) at the 3′ end of exosomal miRNAs, named EXO sequence and extra-seed sequence (hEXO), respectively, have been identified. The EXO and hEXO motifs are recognized by specific proteins that are responsible for miRNA loading within vesicles and control the release of ECmiRNAs [167]. In T cells, the sumoylated form of the heterogeneous nuclear ribonucleoprotein A2B1 (hnRNPA2B1) recognizes the GGAG sequence in miRNAs [168], while the SYNCRIP protein (Synaptotagmin-binding Cytoplasmic RNA-Interacting Protein) binds the GGUC motif in miRNAs from hepatocytes [169]. Another regulator of exosomal miRNA release is the ceramide, a complex lipid highly concentrated at the plasma membrane. The reduction of ceramide biosynthesis through the inhibition of sphingomyelinase-2 (n-Mase2) enzyme in HEK293 cells resulted in decreased exosomal miRNAs, without modifications of the cellular miRNAs concentration [170]. The identification of conserved sequences recognized by specific RNA-binding proteins and the influence of cell membrane composition on ECmiRNAs release support the hypothesis that ECmiRNAs are generated as a result of a well-coordinated process and not as a byproduct of cellular metabolism.

MiRNAs can be also released in extracellular fluids in association with proteins, even though only few studies have shown a non-vesicular origin of ECmiRNAs. Protein-associated ECmiRNAs from both blood plasma and MCF7 cell culture media were found associated with AGO2, a component of the RISC complex [171]. From fractioned human plasma of different healthy donors, several miRNAs (e.g., miR-16, miR-92a, miR-122, miR-150, and miR-142-3p) were immunoprecipitated with AGO2, which harboured them from serum RNAse activity, suggesting the presence of at least two ECmiRNAs populations: i) miRNAs associated to vesicles and ii) miRNAs bound to proteins [172]. Moreover, miRNAs can be loaded on HDLs, which can protect them from the degradation mediated by serum nucleases and prevents the activation of an immune response, allowing the delivery to target cells and their internalization through the plasma membrane. For example, miR-223 can be delivered by HDL to endothelial cells, where, once incorporated within the cell, it retains its biological properties, as indicated by the reduction of the expression of Intracellular Adhesion Molecule-1 (ICAM-1) [173].

The regulation of cell–cell communication through ECmiRNAs is critical in fueling the dynamics and interactions that take place within the tumor microenvironment. For example, cancer cells release ECmiRNAs to “educate” the immune system to spare them from active killing. MiR-24-3p, miR-891a, miR-106a-5p, miR-20a-5p, and miR-1908 were identified in exosomes released by nasopharyngeal cancer cells both in patients’ sera and in vitro models. These miRNAs decreased the MARK1 (Microtubule Affinity Regulating Kinase 1) signalling pathway, resulting in low phosphorylation of ERK (Extracellular-signal Regulated Kinase), STAT1, and STAT3 and increased phosphorylation of STAT5 (Signal Transducer and Activator of Transcription 1, 3, and 5, respectively). As a consequence, T cell proliferation and Th1 and Th17 differentiation were inhibited while inducing commitment towards Treg phenotypes [174]. In hepatocellular carcinoma, non-small-cell lung cancer, pancreatic cancer, and breast cancer, high levels of miR-214 within microvesicles can be detected. Analysis of Lewis lung carcinoma cells implanted in syngeneic recipient mice resulted in increased IL-10 (Interleukin 10) secretion by Treg cells and tumor expansion due to PTEN downregulation mediated by miR-214 [175]. Microvesicles containing TGF-β and miR-23a are released by the IGR-Heu (NSCLC cell line) and K562 (Leukemia cell line) cell lines, and their combined actions inhibits NK-cell activation [176]. Inhibition of the immune response was also observed in the pathogenesis of pancreatic cancer, where exosomal miR-212-3p and miR-203 were transferred to dendritic cells inducing the downregulation of Regulatory factor-X Associated Protein (RFXAP) and TLR4 (Toll-Like Receptor 4), respectively, leading to suppression of the immune response of dendritic cells [177,178]. Another immunomodulatory mechanism is mediated by miR-145: it can be released in microvesicles from colorectal cancer cells to be transferred to tumor-associated macrophages (TAMs), where it downregulates the expression of Histone Deacetylase 11 (HDAC11), promoting their differentiation towards M2-like phenotypes and facilitating tumor progression [179]. MiRNAs can also be released by TAMs to promote tumor growth and expansion. IL-4 (Interleukin 4)-activated TAMs release exosomal miR-223, which favored invasion and aggressiveness in breast cancer cells by the activation of the Mef2c-βcatenin pathway (Myocyte enhancer factor 2c-βcatenin) [180].

ECmiRNAs have been involved in the regulation of angiogenesis and cellular metabolism within the tumor microenvironment. For example, exosomal miR-210-3p (miR-210) was detected in high levels in serum isolated from patients diagnosed with hepatocellular carcinoma. Both in vitro and in vivo studies showed induction of tubulogenesis in endothelial cells by miR-210-3p through inhibition of SMAD4 (Mothers Against Decaplentaplegic homolog 4) and STAT6 (Signal Transducer and Activator of Transcription 6) expression [181]. In a model of lung cancer, secretion of exosomal miR-21 favored angiogenesis, increasing the production of VEGF (Vascular Endothelial Growth Factor) by the surrounding Human Umbilical Vein Endothelial Cells (HUVEC) [182]. Reprogramming of energy metabolism in adipocytes was observed upon secretion of miR-155, miR-126, and miR-144 by breast cancer cells. Specifically, miR-155 and miR-144 promote the beige/brown transition through downregulation of PPARγ pathway (Peroxisome Proliferatored Receptor γ), while miR-126 increases cellular metabolism by repression of IRS1 [183,184,185].

Despite their well-established pro-tumorigenic role, multiple studies suggested that miRNAs can mediate tumor-suppressive function and can counteract tumor progression. Adipocytes stem cells (ASCs) secrete exosomes containing “anti-tumoral” miRNAs, able to inhibit the growth of cancer stem cells (miR-503-3p) and prostate cancer cells (miR-145) [186,187]. This could be potentially relevant from a therapeutic perspective, particularly considering that the release of miRNAs can be induced by stimulation with specific molecules. In 3T3-L1 pre-adipocytes and Ductal Carcinoma In Situ (DCIS) co-culture system, the administration of the antitumor compound Shikonin induced the secretion of miR-140-containing exosomes by 3T3L1 cells. Exosomal miR-140 downregulated the SRY-Box transcription factor 9 (SOX9) in Ductal Carcinoma In situ cells (DCIS), interfering with tumor growth, survival, and invasion [188].

The examples described above represent only a fraction of miRNAs that are involved in mediating cell–cell communication within the tumor microenvironment but are sufficient to highlight the complexity of cancer niche and to give a glimpse of the regulatory networks hijacked by tumor cells at multiple levels to sustain cell transformation and tumorigenesis.

Despite an increasing number of studies confirming the contribution of miRNAs (particularly ECmiRNA) in the development and maintenance of tumor microenvironment, the molecular mechanisms regulating ECmiRNA release and uptake to promote cell–cell communication have not been fully elucidated. More in-depth analyses are required to disclose genetic and molecular mechanisms employed by ECmiRNAs to sustain tumorigenesis and to favor the evolution of tumor cells and of the tumor microenvironment.

MiRNAs and pathways involved in cell–cell communication are summarized in Table 2 and Figure 1.

## 6. miRNAs-Based Therapy and Clinical Trials

miRNAs can be considered biological messengers that transfer and deliver critical information to target cells. Given their ability to induce specific signalling pathways and their stability within extracellular fluids (e.g., ECmiRNAs), they can be used both as tools or targets for anticancer therapy [189].

Restoring physiologic expression of miRNAs that are deregulated in different cancer models has shown promising results. Systemic administration of miR-26a in a mouse model of hepatocellular carcinoma, where expression of miR-26a is generally suppressed, induced G1 cell-cycle arrest in tumor cells, with no adverse effects on the animal. In this model, expression of miR-26 was reduced in cancer cells and administration of recombinant miR-26a via adeno-associated virus was able to increase miR-26a levels and to interfere with tumor growth [190]. Indeed, NK cell-derived exosomes or nanoparticles could delivery miR-186 in neuroblastoma cells, where it was generally downregulated. The restoration of miR-186 statements reduced the expression of oncogenic proteins (e.g., AURKA (Aurora Kinase A) and MYCN (N-Myc) proto-oncogene) and reestablished the immune abilities of NK cells [191].

Recently, antisense oligonucleotides (anti-miRNAs) have been developed to target endogenous miRNAs, reducing their expression/activity and inhibiting tumorigenesis. A cholesterol-modified isoform of anti-miR-221 significantly decreased miR-221 expression in liver cells from mice with advanced hepatocellular carcinoma. After a week of administration, anti-miR-221 accumulated in the tumor cells in vivo, reducing tumor proliferation and increasing overall survival in treated animals [192]. Another example is represented by the anti-miR-17, which suppresses miR-17 expression in kidney cysts, reducing their growth through indirect induction of Pdk1 and Pdk2 (Pyruvate dehydrogenase kinase isoform 1 and 2, respectively), thus demonstrating a promising therapeutic strategy for the treatment of Autosomal Dominant Polycystic Kidney Disease (ADPKD) [193].

In addition to a direct interference with the expression of miRNAs, new pharmacological treatments have been developed to alter cell–cell communication within the tumor microenvironment, inhibiting the release or uptake of miRNA-containing vesicles (e.g., exosomes). For example, knockdown of Rab27 (a GTPase protein involved in the secretory pathway) by short-hairpin RNAs decreased secretion of exosomal miR-494 in melanoma cells. The accumulation of miR-494 within the cells, as a consequence of the reduced release, suppressed tumor growth and metastasis, increasing Bcl2 expression [194]. Alternatively, blocking exosomes uptake using inhibitors of endocytosis hampered the communication between multiple myeloma cells and bone marrow stromal cells, reducing cancer proliferation and drug resistance in vitro [195].

Despite being still in an early phase, some ongoing clinical trials are investigating the possibility to interfere with miRNAs homeostasis as therapeutic strategy (*clinicaltrials.gov* database) [196]. Several phase 2 studies on the anti-miR-122 (*Miravirsen*) for the treatment of Hepatitic C Virus (HCV) infections have been completed (NCT01200420, NCT02508090, and NCT02452814). *Miravirsen* is a Lock-Nucleic Acid (LNA) complementary to miR-122, a liver-specific miRNA involved in HCV pathogenesis [197]. While two phase-1 trials (NCT01829971 and NCT02862145: miR-RX34 Liposomal injection in patients with melanoma and liver cancer, respectively) for cancer treatment have been terminated upon adverse events, the *MesomiR-1* phase 1 study has been completed with promising results in patients with malignant pleural mesothelioma and non-small-cell lung cancer (NCT02369198). *MesomiR-1* is based on a new technology termed “TargomiR”, which uses targeted minicells containing a miRNA mimetic. In particular, in *MesomiR-1,* the TargomiR is composed of the following three elements: (i) a synthetic double-stranded RNA molecule which mimics the tumor suppressor miR-16 [198]; (ii) bacterial vesicles for the delivery of the RNA molecule; and (iii) an anti-Epidermal Growth Factor Receptor (EGFR) antibody which recognizes the EGFR expressed on only cancer cells. *Cobomarsen* (also termed MRG-106), an LNA-modified oligonucleotide inhibitor of miR-155, is another anti-miR that recently entered clinical trial (NCT03837457) for the treatment of patients with mycosis fungoides, the most widespread type of cutaneous T-cell lymphoma [199].

Notwithstanding the great interest for miRNAs as new tools or targets for anticancer therapy, their off-target effects must be taken into account [200]. First, miRNA acts in mRNA repression/degradation by binding the complementary MRE site on the mRNA target [53]. However, an miRNA can bind and suppress other mRNAs in addition to its specific target due to homology of the nucleotide sequence [200]. A perfect complementarity between the nucleotide positions 2–8 or 2–7 of miRNA (-5p or -3p) and MRE [201] and the sequence composition (e.g., an AU enrichment can increase the off-target effects) [202] are important characteristics to consider during the miRNA design in order to reduce the off-target effects. On the other end, the presence of exogenous miRNAs and the composition of its delivery system can induce the activation of the immune response as a non-expected effect [200]. Exogenous single-stranded RNAs are recognized by Toll-like receptors TLRs (e.g., TLR7 and TLR8) of the immune cells, stimulating the expression of interferon (IFN) cytokines [203]. Finally, exogenous miRNAs can saturate the miRNA biogenesis complexes, competing with the endogenous miRNAs and altering the physiological cellular functions. In HeLa cells, the transfections of several miRNAs (e.g., miR-16, miR-124, miR-181, and let-7b) downregulated the expression of their target genes, while an increased expression of miRNA non targeted genes was evaluated, suggesting a competition and saturation of the miRNA machinery [204]. Conjugating miRNA to other molecules (such as lipids, polymers, and antibodies), an extensive study of both its nucleotide composition and its useful concentration can presumably reduce the immune system activation and can ensure the delivery of miRNA to its target cells [200].

## 7. Conclusions

The tumor microenvironment is an organized system where different cell types create a permissive niche where cancer cells can proliferate, survive, and evolve to efficiently evade the immune system. Altering this cell–cell communication network represents a promising strategy for the treatment of cancer.

MiRNAs play a pivotal role in the intercellular communication within the tumor microenvironment. Their transcription is regulated by both intracellular and extracellular factors, and once expressed, miRNA can either act locally within different cellular compartments or, alternatively, they can be released in the extracellular microenvironment as paracrine factors that are up-taken by neighboring cells, where they can regulate several biological processes. The restoration of the balance between oncogenic/oncosuppressor elements (e.g., miRNAs) is one of the main goals for the development of efficient anticancer therapies. In this view, different approaches can be applied. Important efforts have been dedicated to the development of pharmacological treatments that can reduce the expression of oncogenic miRNAs or, conversely, that can activate oncosuppressor miRNAs in order to restore physiologic activity of cellular pathways and, eventually, to inhibit tumorigenesis. In this view, following the observation that ECmiRNAs are extremely stable in the plasma, exogenous synthetic miRNAs can be enclosed within synthetic membrane vesicles to be injected into receiving patients for the treatment of various pathologic conditions characterized by lack of miRNAs (or by their altered metabolism).

Anticancer treatments should focus not only on tumor cells but also on additional elements contributing to the establishment of tumor microenvironment, including ECM, immune cells, and fibroblasts, which represent essential targets to design a more comprehensive and efficient therapeutic approach. Indeed, all the components of the tumor microenvironment cooperate with each other and with the cancer cells. In this context, released miRNAs act as extracellular messengers mediating an active exchange of information and communication between different cells to coordinate homeostatic changes and to shape the microenvironment in response to external cues and metabolic requirements to finally contribute in the establishment of a permissive niche where tumor cells can efficiently grow.

Further studies will be necessary to understand the mechanisms driving both ECmiRNAs secretion and uptake and how they direct and/or influence cell–cell communication. For example, it will be interesting to elucidate how a released miRNA is able to specifically reach its cellular target within the complex tumor microenvironment. In this situation, a more complex ECmiRNAs uptake system than the random endocytosis, micropinocytosis, or phagocytosis should be hypothesized [205,206]. The expression of specific molecules on the surface of membrane vesicles, such as glycolipids or lipoproteins, could facilitate the binding to specific receptors on the membrane of the targeted cells. In a recent study, exosomal miR-21 and miR-29a secreted by lung tumor cells bound the murine TLR7 and the human TLR8 receptors (Toll-Like Receptor 7 and 8, respectively) on macrophages, increasing the secretion of prometastatic inflammatory cytokines both in in vitro and in vivo experiments [207].

Alternatively, ECmiRNAs are secreted within the biological fluid and could potentially be up-taken by all the cells in the niche. Following their entry in the target cells, genes regulated by the ECmiRNAs could be different among different targeted cells, resulting in a highly heterogeneous and complex response. Whether ECmiRNAs scan and search for their specific target cells or whether there is no selective internalization of ECmiRNAs remain open questions. Alternatively, only a specific putative cellular target could uptake the released ECmiRNAs, activating a certain cellular response. A better comprehension of cell–cell communication will be useful to identify the putative “weak link” within the complex and well-developed tumor microenvironment. Therefore, a better characterization of the molecular mechanisms promoting secretion/uptake of miRNAs and the identification of the role of miRNAs in cell–cell communication will provide alternative approaches and promising targets for the development of novel anticancer therapies.

## Figures and Tables

**Figure 1 cells-09-00220-f001:**
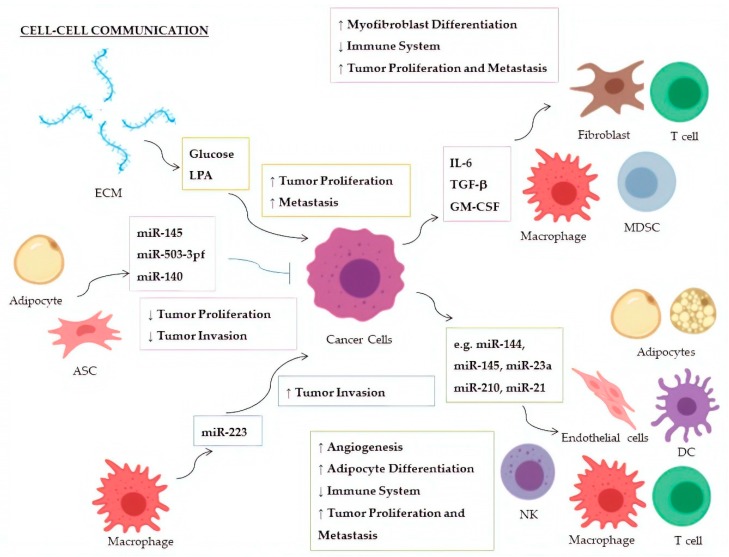
Cell–cell communication within the tumor microenvironment: On the left, factors and ECmiRNAs released by cells or extracellular matrix toward cancer cells; on the right, factors and ECmiRNAs released by cancer cells that stimulate biological pathways in recipient cells. ↑ Upregulation; ↓ Downregulation; ASC: Adipose-derived Stem Cell; DC: Dendritic Cell; ECM: Extracellular Matrix; LPA: Lysophosphatidic Acid; MDSC: Myeloid-derived suppressor cell; NK: Natural Killer cell.

**Table 1 cells-09-00220-t001:** Examples of induced miRNAs by cell–cell communication within the tumor microenvironment.

Factor	miRNA	Donor	Recipient	Targeted Molecule	Target Expression	Effect	Ref.
**Glucose**	miR-451	CSPGs (ECM)	Glioblastoma	AMPK	↓	Increased cellular proliferation; decreased tumor migration	[147,148]
**LPA**	miR-21	ECM	Breast Cancer	PTEN	↓	Promotion of metastasis	[149]
**IL-6**	miR-19a-3p	Breast Cancer	TAMs	Fra-1	↑	Increased expression of VEGF and STAT3; M2-polarization of TAMs; increased tumor metastasis and migration	[150]
**TGF-** **β**	miR-491	Colorectal Cancer	CD8^+^ T cells	CDK4, TCF-1, Bcl-xL	↓	Decreased cell proliferation and IFN-γ production; increased apoptosis	[151]
**TGF-** **β1**	miR-494	Breast Cancer	MDSCs	PTEN	↓	Accumulation of MDSCs in tumor tissues; Akt activation; tumor growth and metastasis	[152]
**TGF-** **β**	miR-21	Colorectal Cancer	CAFs	α−SMAD	↑	Differentiation to myofibroblasts; increased tumor cell proliferation	[153]
**TGF-** **β**	miR-143	Gastric Cancer	CAFs	Collagen type III	↑	Promotion of tumorigenesis	[154]
**GM-CSF**	miR-200c	Bone marrow exposed to several cancers	MDSCs	PTEN, FOG2	↓	Expansion of MDSCs in tumor tissues and suppression of immune system; PI3K/Akt, STAT3 activation	[155]

↑ Upregulation; ↓ Downregulation. ECM: Extra-Cellular Matrix; CSPG: Glycosylated Chondroitin Sulfate Proteoglycans; LPA: Lysophosphatidic Acid; TCF-1: Transcription Factor 1; Bcl-xL: B-cell lymphoma-extra Large; IFN-γ: Interferon γ; TAMs: Tumor-Associated Macrophages; STAT3: Signal Transducer and Activator of Transcription 3; MDSCs: Myeloid-derived Suppressor Cells; CAFs: Cancer-Associated Fibroblasts.

**Table 2 cells-09-00220-t002:** Examples of exosomal miRNAs involved in cell–cell communication within the tumor microenvironment.

miRNA	Donor	Recipient	Targeted Molecule	Target Expression	Effect	Ref.
**miR-24-3p, miR-891a, miR-106a-5p, miR-20a-5p, miR-1908**	NPCs	Th1 Cells	MARK1	↑	Increase of the pro-inflammatory cytokines IL-1β, IL-6, and IL-10; reduction of INF-γ, IL-2, and IL-17; inhibition of T-cell proliferation and Th1, and Th17 differentiation; promotion of Treg proliferation	[174]
**miR-214**	Lung Cancer	Treg Cells	PTEN	↓	Tregs expansion; increase of IL-10; tumor growth	[175]
**miR-23a**	Leukemia, Lung Cancer	NKs	INF-γ, CD107a	↓	Decrease of immunosuppressive action by NK cells	[176]
**miR-212-3p**	Pancreatic Cancer	DCs	RFXAP	↓	Decreased expression of MHC-II; induction of DCs immune tolerance	[177]
**miR-203**	Pancreatic Cancer	DCs	TLR4	↓	Decrease of TNF-α and IL-12 production	[178]
**miR-145**	Colorectal Cancer	TAMs	HDAC11, TLR4	↓	Promotion of TMAs polarization to M2-like macrophages; promotion of tumor progression and enlargement of tumor volume	[179]
**miR-223**	TAMs	Breast Cancer	Mef2c-β-catenin	↑	Promotion of tumor invasion and aggressiveness	[180]
**miR-210-3p (miR-210)**	Liver Cancer	CAEs	SMAD4, STAT6	↓	Enhanced angiogenesis; tubulogenesis promotion of endothelial cells	[181]
**miR-21**	Lung Cancer	Endothelial Cells	STAT3	↑	Enhanced VEGF expression and angiogenesis	[182]
**miR-155**	Breast Cancer	Adipocytes	PPARγ	↓	Promotion of beige/brown adipocyte differentiation and remodelling of metabolism	[183]
**miR-126a**	Breast Cancer	Adipocytes	IRS1	↓	Increased catabolism and release of metabolites and HIF1α expression in adipocytes	[184]
**miR-144**	Breast Cancer	Adipocytes	MAP3K8/ ERK1/2/ PPARγ	↓	Promotion of beige/brown adipocyte differentiation	[184]
**miR-145**	ASCs	Prostate Cancer	Bcl-xL	↓	Promotion of apoptosis; decrease of tumor growth	[186]
**miR-503-3pf**	ASCs	Liver Cancer	Pluripotency genes (e.g., Nanog)	↓	Smaller tumor volume and cancer stemness-attenuation	[187]
**miR-140**	Adipocytes	DCIS	SOX9	↓	Inhibition of tumor invasion following antitumor Shikonin treatment	[188]

↑ Upregulation; ↓ Downregulation; NPCs: Nasopharyngeal Cancer cells; NKs: Natural Killer cells; DCs: Dendritic Cells; MHC-II: Major Histocompatibility Complex II; TAMs: Tumor-Associated Macrophages; CAEs: Cancer-Associated Endothelial cells; HIF1α: Hypoxia-Inducible Factor 1α; MAP3K8: Mitogen-Activated Protein Kinase Kinase Kinase 8; ASCs: Adipose-derived Stem Cells; DCIS: Ductal Carcinoma In Situ.

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
