# Peer review of "miRNAs as Influencers of Cell–Cell Communication in Tumor Microenvironment"

_cells, 2020, doi:10.3390/cells9010220_

Round 1

Reviewer 1 Report

In this paper, Conti et al. Reviewed the role of extracellular miRNAs in tumor cell biology as well as how cancer cells can release miRNAs that may impact cáncer development. Additionally, the authors reviwed therapeutics approaches base don miRNA tolos. The review is of high interest both for clinicians and basic investigators. It is very well written and documented. Some minor points should be reviewed.

In page 2,4th paragraph, the authors state that miRNAs may be released in the “extracelular matrix”. This sentence creates confusion and authors should better use the term “circulation” or “blood”. Page 3, 3rd paragraph. The authors used the term miRNA*, the nomenclature has changed as the authors may know (-3p and -5p), in addition many -3p miRNAs (*) have important roles in physiology and pathophysiology, thus this paragraph should be arranged concordantly. First paragraph of page 6 about therapeutics on v-miRNA is of interest and should be siupported by at least one reference. In page 6, the authors state “Over 2000 human mature miRNAs have been identified”, additional details should be given (human miRNAs described in the last version of miRBase). Page 7 last paragraph. PTEN has already been described, only the acronym is needed. Indeed, authors should place acronym after the name (page 7, 1st paragraph). Some proteins are not acronymized, why? Some typo errors should be checked (p.8 MDA-MB-231 instead of MDMA-MB-231; p.13 miR-503-3pf?) In Table 1 and in the text page 9, I would suggest to put together the different references concerning TGF-b Page 12. The authors should discuss results of Arroyo et al PNAS 2011, about the transport of miRNAs in plasma and serum by Ago2. Some words on potential off-target effects of miRNA therapeutics should included in chapter 6

Author Response

Corrections in the Word manuscript file have been reported in Red.

Answers to the Reviewer have been reported in Bold below.

Reviewer 1

Comments and Suggestions for Authors:

In this paper, Conti et al. Reviewed the role of extracellular miRNAs in tumor cell biology as well as how cancer cells can release miRNAs that may impact cancer development. Additionally, the authors reviewed therapeutics approaches based on miRNA tolls. The review is of high interest both for clinicians and basic investigators. It is very well written and documented. Some minor points should be reviewed.

1) In page 2,4th paragraph, the authors state that miRNAs may be released in the “extracellular matrix”. This sentence creates confusion and authors should better use the term “circulation” or “blood”.

Our answer to point 1:

We have changed the sentence “extracellular matrix” with the terms “extracellular circulation and blood” at page 2, line 69.

2) Page 3, 3rd paragraph. The authors used the term miRNA*, the nomenclature has changed as the authors may know (-3p and -5p), in addition many -3p miRNAs (*) have important roles in physiology and pathophysiology, thus this paragraph should be arranged concordantly.

Our answer to point 2:

We have changed the nomenclature of miRNAs using -3p and -5p at page 3, lines 124-125. The paragraph has been arranged underlying the important role of -3p miRNAs in physiology and pathophysiology, introducing an example at page 3, lines 128-131 with related reference n. 43.

3) First paragraph of page 6 about therapeutics on v-miRNA is of interest and should be supported by at least one reference.

Our answer to point 3:

We have introduced the references 96 and 98 to support the sentences of therapeutics on v-miRNAs at page 6, line 283.

4) In page 6, the authors state “Over 2000 human mature miRNAs have been identified”, additional details should be given (human miRNAs described in the last version of miRBase).

Our answer to point 4:

We have detailed the sentence adding “human miRNAs described in the last version of miRBase” at page 6, line 289.

5) Page 7 last paragraph. PTEN has already been described, only the acronym is needed.

Our answer to point 5:

We have deleted the description of PTEN and left its acronym at page 8, line 393.

6) Indeed, authors should place acronym after the name (page 7, 1st paragraph). Some proteins are not acronymized, why?

Our answer to point 6:

We have placed the acronym after the name of the protein, at page 7, line 315. Indeed, we have checked the whole manuscript introducing the acronyms or the extended names when absent.

7) Some typo errors should be checked (p.8 MDA-MB-231 instead of MDMA-MB-231; p.13 miR-503-3pf?)

Our answer to point 7:

We have checked the whole text and deleted typo errors: MDA-MB-231 at page 9, lines 438-439; miR-503-3p, at page 14 line 641.

8) In Table 1 and in the text page 9, I would suggest to put together the different references concerning TGF-b Page 12.

Our answer to point 8:

We have put together the different references concerning TGF-b both in the text arranging the paragraphs (lines 489-510 at pages 10-11) and in the table 1 at page 11.

9) The authors should discuss results of Arroyo et al PNAS 2011, about the transport of miRNAs in plasma and serum by Ago2.

Our answer to point 9:

We discussed the results of Arroyo J. D. et al PNAS 2011 about the transport of miRNAs in plasma and serum by Ago2 in chapter 5 at page 13, lines 589-592 with related reference n. 172.

10) Some words on potential off-target effects of miRNA therapeutics should include in chapter 6.

Our answer to point 10:

We discussed the potential off-target effects of miRNA therapeutics in chapter 6 at page 18, lines 717-735 with related references n. 53, 200, 201, 202, 203, 204.

Reviewer 2 Report

The authors propose to review the role of miRNAs as cell-cell communication influencers in tumor microenvironment.

Actually, from reading the manuscript my feeling is that not that much particular attention is given to cancer, maybe a broader and less specific title would suit better the manuscript.

Some minor typos or grammar corrections are needed throughout the manuscript.

The authors provide a good overview of miRNAs origin, biogenesis, and regulatory functions. Further, they review nicely the role of miRNAs in some relevant diseases such as AD, obesity, and cancer. However, I think that the manuscript would be even more valuable if the authors would try to review also the involvement of miRNAs-mitochondria axis (in the particular diseases they have chosen). Given that mitochondria are much implicated in these diseases, some references have already approached the involvement of miRNAs in mitochondrial regulation in these contexts (see refs. below).

Further the authors review very elegantly the role of miRNAs in the tumor microenvironment, the extracellular miRNAs as cell-cell comunicators, and also some miRNA-based therapies.

DOI 10.2174/0929867322666150514095910 (for the role of miRNAs in mitochondria in metabolism)

DOI 10.1007/978-3-319-22671-2_8 (for the role of miRNAs in mitochondria in cancer, metabolism, and AD)

DOI 10.3390/genes5040865 (the role of miRNAs in mitochondria in metabolism and cancer)

DOI 10.1016/j.omtn.2019.10.016 (role of miRNAs in mitochondria in cancer stem cells)

Author Response

Corrections in the Word manuscript file have been reported in Red.

Answers to the Reviewer have been reported in Bold below.

Reviewer 2

Comments and Suggestions for Authors:

1) The authors propose to review the role of miRNAs as cell-cell communication influencers in tumor microenvironment.

Actually, from reading the manuscript my feeling is that not that much particular attention is given to cancer, maybe a broader and less specific title would suit better the manuscript.

Our answer to point 1:

At present the manuscript consists of 18 pages plus references.

The content of more than 10 pages are devoted to cancer views under different perspectives.

However, the other 8 pages are not fully lacking of cancer references.

Therefore, we do believe that the present title could be maintained as appropriate.

2) Some minor typos or grammar corrections are needed throughout the manuscript.

Our answer to point 2:

We have checked the whole manuscript correcting typos and grammar errors.

3) The authors provide a good overview of miRNAs origin, biogenesis, and regulatory functions. Further, they review nicely the role of miRNAs in some relevant diseases such as AD, obesity, and cancer. However, I think that the manuscript would be even more valuable if the authors would try to review also the involvement of miRNAs-mitochondria axis (in the particular diseases they have chosen). Given that mitochondria are much implicated in these diseases, some references have already approached the involvement of miRNAs in mitochondrial regulation in these contexts (see refs. below).

Further the authors review very elegantly the role of miRNAs in the tumor microenvironment, the extracellular miRNAs as cell-cell communicators, and also some miRNA-based therapies.

3.1) DOI 10.2174/0929867322666150514095910 (for the role of miRNAs in mitochondria in metabolism) Now corresponding to reference number 87.

Our answer to point 3.1:

We have discussed the involvement of miRNAs in mitochondrial regulation introducing a general overview of the role of miRNAs in mitochondria in chapter 2 at pages 5-6, lines 246-262 with related references n. 87 and 88 as suggested by the Reviewer plus references n. 89, 90 and 91.

3.2) DOI 10.1007/978-3-319-22671-2_8 (for the role of miRNAs in mitochondria in cancer, metabolism, and AD) Now corresponding to reference number 88.

Our answer to point 3.2:

We have discussed the involvement of the miRNAs-mitochondrial axis for the development of Alzheimer’s Disease (AD) at page 7, lines 332-339 with related reference n. 88 as suggested by the Reviewer plus references n. 110 and 111.

3.3) DOI 10.3390/genes5040865 (the role of miRNAs in mitochondria in metabolism and cancer) Now corresponding to reference number 124.

Our answer to point 3.3:

We have discussed the involvement of the miRNAs-mitochondrial axis for the development of Obesity at page 8, lines 375-380 with related reference n. 124 as suggested by the Reviewer plus reference n. 125.

3.4) DOI 10.1016/j.omtn.2019.10.016 (role of miRNAs in mitochondria in cancer stem cells) Now corresponding to reference number 144.

Our answer to point 3.4:

We have discussed the involvement of the miRNAs-mitochondrial axis for the development of Cancer at pages 9-10, lines 445-457 with related references n. 124 and 144 as suggested by the Reviewer plus reference n. 143.